# MALT: Improving Reasoning with Multi-Agent LLM Training

**Sumeet Ramesh Motwani**[1], **Chandler Smith**[2], **Rocktim Jyoti Das**[3], **Rafael Rafailov**[4],
**Ivan Laptev**[3], **Philip Torr**[1], **Fabio Pizzati**[3], **Ronald Clark**[1], **Christian Schroeder de Witt**[1]

[1]University of Oxford   [2]Cooperative AI Foundation   [3]MBZUAI   [4]Stanford University

Correspondence: sumeet.motwani@eng.ox.ac.uk, cs@robots.ox.ac.uk
Project page: https://multiagentllmtraining.com/

## Abstract

Large Language Models (LLMs) often produce answers with a single chain-of-thought, which restricts their ability to explore reasoning paths or self-correct flawed outputs in complex tasks. In this paper, we introduce MALT (Multi-Agent LLM Training), a novel post-training strategy that divides the reasoning process into generation, verification, and refinement steps using a sequential pipeline of heterogeneous agents. During data generation, each agent is repeatedly sampled to form a multi-agent search tree, where final outputs are graded against ground-truth data. We then apply value iteration to propagate reward signals back to each role-conditioned model, automatically producing multi-agent post-training data without human or teacher-model supervision. Our off-policy approach allows each agent to specialize by learning from correct and incorrect trajectories, ultimately improving the end-to-end reasoning chain. On MATH, GSM8K, and CSQA, MALT surpasses the same baseline LLM with relative improvements of 15.66%, 7.42%, and 9.40%. It also generalizes to more challenging benchmarks, marking an early advance in multi-agent cooperative training.

## 1 Introduction

Reasoning with Large Language Models (LLMs) is inherently challenging, particularly for tasks that require multi-step deductions, intermediate computations, or self-correction (Xiang et al., 2025). Recent work on multi-agent approaches—such as debate frameworks (Du et al., 2024) or orchestrated problem-solving (Fourney et al., 2024)—has shown promise by assigning different parts of the reasoning process to separate models, allowing for refinement and increased deliberation at inference time (Snell et al., 2024). However, the underlying LLMs are typically the same base model pre-trained on data that lacks exposure to the specialized roles or meta-strategies involved in solving complex problems, which introduces a distribution shift while reasoning at test-time (Xiang et al., 2025; Han et al., 2024). An open gap thus persists: *How can we jointly train LLMs to specialize in a multi-agent setting to improve reasoning?* Such a gap persists due to several key obstacles: First, in an end-to-end supervised training approach, it is difficult to propagate gradient signals through multiple discrete token-outputs. Second, there is limited role-specific labeled training data. Third, credit assignment is difficult in reinforcement learning with only sparse outcome rewards (Tumer & Agogino, 2007). Finally, it is important to determine what type of multi-agent setup is even useful to implement meta-strategies that can improve reasoning so that more inference compute can be spent efficiently.

In this paper, we address these challenges by introducing an intuitive strategy to jointly post-train specialized LLMs in a generate-verify-refine pipeline. This is analogous to how humans tackle complex tasks—drafting an initial answer, thoroughly verifying and critiquing it, and refining the solution to match their specifications (Qian et al., 2024). We propose a method that automatically generates large-scale, labeled data for each agent via a multi-agent credit assignment approach. With this dataset, our post-training approach

enables role-conditioned models to learn from both positive and negative reasoning trajectories—providing a path to improve reasoning performance across a range of problems with trained multi-agent setups. We aim to address this critical gap with Multi-Agent LLM Training (MALT), a new post-training method that we apply to three models (a generator, verifier, and refiner) solving complex reasoning problems together.

MALT employs a sampling procedure that expands a search tree based on each model's outputs with an exponential branching factor, thereby producing large amounts of useful synthetic data. Role-specific data, particularly when augmented with rationales, has been shown in previous work to significantly improve performance (Zelikman et al., 2022). However, this approach leads to a credit assignment problem where internal reasoning branches may be correct or incorrect and must be labeled solely based on final outcome rewards to post-train models. To address this, we propose a value-iteration-based attribution strategy (Sutton & Barto, 2018). By analyzing only the outputs of the search tree, our method identifies which model introduced an error, enabling credit assignment without requiring additional multi-agent training data or an oracle policy. This eliminates the need for intervention in selecting trajectories, generating role-specific data, or designing value functions, and instead automatically produces reasoning traces from the search tree for post-training via supervised fine-tuning and preference optimization (Rafailov et al., 2023). MALT integrates all these steps into an intuitive joint-training procedure, providing performance improvements and serving as an early step toward unifying search and learning for multi-agent LLM systems. Our contributions are as follows:

- We are the first to introduce Multi-Agent LLM Training (MALT) to cooperatively post-train a specialized generator, verifier, and refinement model on challenging reasoning tasks by leveraging search and subsequent fine-tuning.
- We propose a search-tree expansion process and use a value iteration technique to propagate outcome rewards to automatically attribute correct and incorrect reasoning traces to individual agents. This synthetic data can be utilized for supervised fine-tuning and reinforcement learning in general multi-agent reasoning pipelines.
- We apply MALT to math and common sense reasoning questions from MATH, CSQA, and GSM8K, obtaining a relative boost of 15.66%, 9.40%, and 7.42% over a single-model baseline, also providing other comparisons and ablations.
- We show how MALT on the benchmarks discussed above generalizes to far more challenging reasoning tasks such as GSM-Symbolic—where, MALT almost matches the performance of an 8.75× size model from the same series.

## 2  Related Work

**Advanced Reasoning and Inference Compute:** Multi-agent architectures have emerged as an effective way to handle complex reasoning by distributing problem-solving roles among distinct models (Cobbe et al., 2021; Xiang et al., 2025). Frameworks such as AgentQ (Putta et al., 2024) and AutoGen (Wu et al., 2024) produce solutions through guided search and compositional dialogue, but rely on a single underlying model for multiple tasks or omit a dedicated post-trained verification mechanism. Debate-style (Du et al., 2024) and orchestrated (Fourney et al., 2024) multi-agent methods enable increased deliberation at test-time with multiple interacting agents. However, the underlying models lack advanced training for role specification, which may lead to suboptimal performance or distribution shifts during test-time (Xiang et al., 2025). In contrast, single-agent approaches utilize techniques such as introspection (Qu et al., 2024) and self-critique (Saunders et al., 2022; Kumar et al., 2024) to detect their own errors, but they typically lack the mechanisms to correct those errors in a single inference pass (Ye et al., 2024). Unlike others, we aim to learn specialized agents with dedicated functions, thanks to our novel credit assignment strategy and post-training procedure.

**Training Data and Preference Optimization:** Using synthetic data generation with preference-based training has emerged as an important strategy for boosting LLM performance (Zelikman et al., 2022). (Setlur et al., 2024) demonstrate that training on both correct and incorrect synthetic solutions, optimized with Direct Preference Optimization

(Rafailov et al., 2023, DPO), significantly improves math reasoning performance. (Singh et al., 2024) surpass purely human-based approaches on challenging math and coding tasks by repeatedly generating, filtering, and fine-tuning with scalar feedback, while (Pang et al., 2024) demonstrate how preference signals applied across entire chains-of-thought refine intermediate reasoning steps. Our work unifies the aforementioned techniques to create a multi-agent pipeline that orchestrates a generator, verifier, and refinement model, leveraging search, synthetic data generation, and preference-based post-training to enable robust multi-step reasoning. We discuss additional related work in Appendix A.7.

## 3 Preliminaries

**Chain of Thought (CoT)**   An LLM's output for a question $q$, with natural language reasoning steps $(s_1, \ldots, s_n)$ followed by an answer $a$, can be viewed as a CoT process sampled from the distribution:

$$p_d(a \mid q) \propto \int p_d(a \mid s_{1:n}, q) \prod_{t=1}^{n} p_d(s_t \mid s_{<t}, q) \, dS.$$

More generally for complex problems, (Xiang et al., 2025) provide a meta-generalization that describes the true solution-generating process involving extra latent steps $(z_1, \ldots, z_k)$:

$$p_d(a, s_{1:n} \mid q) \propto \int p_d(a, s_{1:n} \mid z_{1:k}, q) \prod_{t=1}^{k} p_d(z_t \mid z_{<t}, q) \, dZ.$$

These meta-variables could capture iterative or corrective steps, unfolding a more adaptive, multi-stage reasoning trajectory. In a multi-agent setup, we consider a process with three specialized models: a Generator $G$, a Verifier $\mathcal{V}$, and a Refinement model $R$ producing $g$, $v$, and $r$ respectively. This can be mapped onto the meta-CoT setting, where

$$p(a \mid q) = \int p_G(g \mid q) \, p_{\mathcal{V}}(v \mid g, q) \, p_R(a \mid g, v, q) \, d(g, v).$$

Here, $g$ and $v$, from the standpoint of standard $\langle q, a \rangle$ datasets, act like latent meta-steps guiding more complex solution processes, akin to $z_t$ in meta-CoT. By explicitly modeling these roles during joint post-training (see Section 4), the multi-agent sequential deliberation process allows us to tackle problems beyond a single-pass CoT setting.

**Supervised Finetuning (SFT)**   Given $\mathcal{D}_{\text{train}}^{\text{SFT}}$ containing positive demonstrations (e.g. correct generator outputs or useful verifier critiques), we can carry out SFT:

$$\mathcal{L}_{\text{SFT}}(\pi_\theta) = -\mathbb{E}_{(x,y) \sim \mathcal{D}_{\text{train}}^{\text{SFT}}} \sum_{t=1}^{T} \log \pi_\theta(y_t | y_{<t}, x).$$

which is a simple and relatively successful technique to improve reasoning performance.

**Direct Preference Optimisation (DPO)**   In SFT, we can exclusively use positive samples in order to improve our test set performance. However, using preference optimization also allows us to leverage negative samples, $y^- \prec y^+$, where $\prec$ means that $y^+$ is preferred over $y^-$. DPO (Rafailov et al., 2023) is one such method—a more efficient alternative to classical RLHF (Christiano et al., 2017)—because instead of employing a full reinforcement learning loop, it directly optimizes the following contrastive objective:

$$\mathcal{L}_{\text{DPO}}(\pi_\theta) = -\mathbb{E}_{(x,y^+,y^-) \sim \mathcal{D}_{\text{train}}^{\text{DPO}}} \sigma \left( \beta \log \frac{\pi_\theta(y^+ \mid x)}{\pi_{\text{ref}}(y^+ \mid x)} - \beta \log \frac{\pi_\theta(y^- \mid x)}{\pi_{\text{ref}}(y^- \mid x)} \right).$$

We assume $\pi_{\text{ref}}$ is a reference policy, obtained through SFT. $\mathcal{D}_{\text{train}}^{\text{DPO}}$ is a dataset of triplets including both inputs and positive/negative outputs, and $\beta$ is the hyperparameter scaling reward signal strength. We provide a theoretical link between optimizing the DPO objective and the optimal RL policy in Appendix A.6.1.

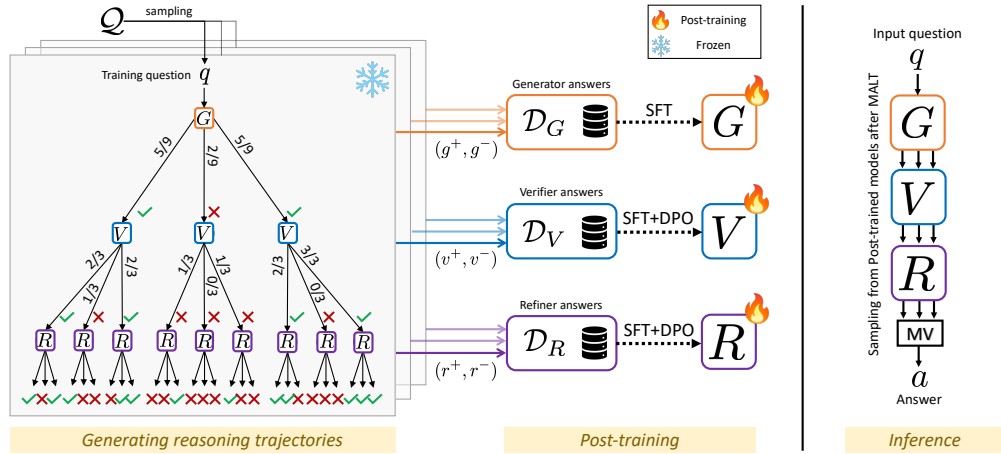

Figure 1: **MALT Method Overview.** Given an input, we consider a three-agent system composed of a Generator for initial answer production, a Verifier providing a critique, and a Refinement Model integrating all intermediate reasoning steps into a final output. For questions in the training set, we introduce a tree search and credit assignment process **(Left)** to generate synthetic datasets with reasoning trajectory preference pairs for each model. These are used to post-train individual models **(Center)**. During inference over the test-set, we perform three parallel sequential passes through the multi-agent setup, and return the final answer obtained via majority voting **(Right)**.

# 4 Method: Multi-Agent LLM Training

In reasoning tasks, a single LLM must handle all aspects of the solution generation process - often leading to limited exploration, a lack of self-correction, and difficulty refining partial steps (Ye et al., 2024). However, the reasoning process can be broken down into a system of decentralized LLMs, where each model has differing objectives and/or partial observability. As described in Section 3, this represents a meta-CoT (Xiang et al., 2025) setting where intermediate outputs in the multi-agent setup can improve the overall reasoning process. Although in fully observable and cooperative cases, systems of LLM agents could technically be simulated by a single centralized LLM, a decomposition into separate heterogeneous LLM agents offers various benefits analogous to those observed in decentralized multi-agent learning (Boutilier, 1996; Schroeder de Witt et al., 2020). Decentralization factorizes large joint action spaces, allowing each agent to focus on smaller sub-tasks under its own partial observability, leading to more targeted exploration and clear credit assignment (Tan, 1993). Here, we present our methodology for a multi-agent setting consisting of a sequential heterogeneous process where agents can be trained based on joint rewards.

## 4.1 Multi-Agent Inference Setting

We formulate our multi-agent inference setting as a collaborative reasoning framework designed to solve complex tasks. Let $\mathcal{Q}$ denote a dataset of natural language questions, where each $q \in \mathcal{Q}$ represents a specific task instance. The objective is to generate a prediction $a \in \mathcal{A}$ for a given input $q$, where $\mathcal{A}$ is the set of all possible answers. During training, there exists a ground truth function $f : \mathcal{Q} \to \mathcal{A}$, where $f(q) = a^{\text{GT}}$ serves as the ideal prediction for evaluating $a$. Our framework consists of three specialized LLMs acting as distinct agents, each defined as a function:

1. **Generator** ($G : \mathcal{Q} \times \mathcal{P}_G \to \mathcal{O}_G$): Produces an initial response to the question.
2. **Verifier** ($\mathcal{V} : \mathcal{O}_G \times \mathcal{Q} \times \mathcal{P}_{\mathcal{V}} \to \mathcal{O}_{\mathcal{V}}$): Critiques the generated response for issues.
3. **Refinement Model** ($R : \mathcal{O}_G \times \mathcal{O}_{\mathcal{V}} \times \mathcal{Q} \times \mathcal{P}_R \to \mathcal{O}_R$): Integrates feedback to improve the final prediction.

Here, $\mathcal{P}_G, \mathcal{P}_{\mathcal{V}}, \mathcal{P}_R$ denote the set of prompts for each model, and $\mathcal{O}_G, \mathcal{O}_{\mathcal{V}}$, and $\mathcal{O}_R$ are the sets of possible outputs for the generator, verifier, and refiner respectively. $\mathcal{A}$ is the set of possible answers extracted from the refiner's output with a fixed deterministic function $T$.

Formally, we define the interaction between these agents as :

$$g_o = G(q, p_g) \in \mathcal{O}_G; \; v_o = \mathcal{V}(q, p_v, g_o) \in \mathcal{O}_\mathcal{V}; \; r_o = R(q, p_r, g_o, v_o) \in \mathcal{O}_R; \; a = T(r_o) \in \mathcal{A}.$$

This setup is reminiscent of how LLMs are used in production, where they receive initial prompts containing questions, feedback, and are then asked to refine answers (Wang et al., 2024).We demonstrate in Section 5 that this inference setting enhances performance compared to single-model approaches. The key insight, however, relies on leveraging this multi-agent inference setting to generate synthetic data that scales exponentially with respect to a branching factor. Below, we discuss out data generation and post-training setup.

## 4.2 Collecting Reasoning Trajectories

In standard single-LLM setups, a single model simply produces an answer $a$ for each question $q$. By contrast, our multi-agent framework uses three specialized agents—Generator, Verifier, and Refiner—sequentially. To enable training of these agents, we need to capture how each agent's output contributes to the final prediction and whether the overall solution is correct or incorrect. A reasoning trace leading to $a$ contains $[g_o, v_o, r_o]$, where the multi-agent answer generation process will be:

$$a = T\Big(R\big(q, \; p_r, \; G(q, \; p_g), \; \mathcal{V}(q, \; p_v, \; G(q, \; p_g))\big)\Big).$$

During data generation, $G, \mathcal{V}$, and $R$ all share the same base model parameters; in the subsequent training phase (Section 4.3), each policy is updated independently to specialize in its respective role. An outcome reward function $\mathcal{R} : \mathcal{A} \times \mathcal{A} \rightarrow \{0, 1\}$, based on the ground truth in the training set, evaluates $a$ to mark the trajectory as either correct (1) or incorrect (0). Specifically, for a predicted answer $a$ and ground truth $a^{GT}$, we define $\mathcal{R}(a, a^{GT}) = 1$, if $a = a^{GT}$, and 0 otherwise.

To collect role-specific post-training data, we use $< q, a^{GT} >$ pairs from an initial training set $\mathcal{D}_{train}$ associated with each benchmark. Our solution is illustrated in Figure 1 (left). The following sampling strategy is employed for all models, with a branching factor of $n$. For each problem $q_i \in \mathcal{D}_{train}$ in the training data, we sample $n$ completions $\{g_{i,j} \sim G(q_i)\}_{j=1}$ from the generator. Then, for each $G$ output $g_{i,j}$, we produce $n$ verifications $\{v_{i,j,k} \sim \mathcal{V}(g_{i,j}, q_i)\}_{k=1}^n$. Finally, for each $V$ output $v_{i,j,k}$, we generate $n$ refinements $\{r_{i,j,k,l} \sim R(g_{i,j}, v_{i,j,k}, q_i)\}_{l=1}^n$.

This process results in $n^3$ trajectories for each training example, totaling $|\mathcal{D}_{train}| \cdot n^3$ trajectories. We use the outcome reward model $\mathcal{R}$ to label the refinement outputs as correct ($\checkmark$) or incorrect ($\times$). The exponential branching factor is very useful to collect a large number of diverse training samples (and inference can be parallelized for efficiency while post-training will just rely on the fixed dataset collected).

To effectively utilize reward signals from the refinement model's outputs, we adopt a **value iteration** approach to propagate values backward through the reasoning chain. Specifically, we compute the *expected value* of each intermediate output (i.e. generator and verifier outputs) based on the values of downstream outputs from the refiner. This global pooling approach is useful because an intermediate step's utility depends on how it influences downstream outputs and final correctness (see Figure 1, Left). Partial solutions can still yield correct answers once refined, so simply providing binary labels is insufficient. By assigning each output an expected value, we better capture the influence on the final reward. This also allows MALT to be generalized to settings where intermediate steps may not carry the same labels, and each intermediate output can now be credited in proportion to its impact on the final solution.

**Value Function Definitions**   We define the value of each *refinement* node $r_{i,j,k,l}$ by directly comparing its final answer $a$ to the ground truth:

$$\mathbf{V}(r_{i,j,k,l}) = \mathcal{R}\big(T(r_{i,j,k,l}), a^{GT}\big) \in \{0, 1\}.$$

The value of a $V$ output $v_{i,j,k}$ is then computed as the expected value of its child refinements:

$$\mathbf{V}(v_{i,j,k}) = \mathbb{E}_l\big[\mathbf{V}(r_{i,j,k,l})\big] \approx \frac{1}{n}\sum_{l=1}^{n}\mathbf{V}(r_{i,j,k,l}).$$

Similarly, the value of a $G$ output $g_{i,j}$ is the expected value of its child verifier outputs:

$$\mathbf{V}(g_{i,j}) = \mathbb{E}_k\big[\mathbf{V}(v_{i,j,k})\big] \approx \frac{1}{n}\sum_{k=1}^{n}\mathbf{V}(v_{i,j,k}).$$

Each output state's empirical mean is a *Monte Carlo* approximation (an unbiased sample average) of its true expected correctness, indicated by "$\approx$" for each $q$. This process propagates reward signals from final outputs back through the tree to each intermediate state, capturing each output's overall utility.

**Thresholding and Binarization**   To prepare the data for training (SFT and DPO), we *binarize* values using a threshold of 0.5, aligning with majority voting principles. Nodes with values greater than 0.5 are labeled as correct ($\checkmark$), and those with values less than or equal to 0.5 are labeled as incorrect ($\times$). The intuition behind this strategy is discussed in more detail in Appendix A.6.1, where we demonstrate that it ensures the policy's expected value monotonically increases. Formally, for each output state $s$ (i.e. output from any model in the multi-agent setup), we define its label $\hat{s}$ as $\checkmark$ if $\mathbf{V}(s) > 0.5$, and $\times$ otherwise.

### 4.3   MALT Post-training

With each output assigned a value, the goal is to now use these for post-training the models (see Figure 1, Center). We first detail how training data for each model is generated: Each refinement output $r_{i,j,k,l}$ has an associated value $\mathbf{V}(r_{i,j,k,l}) \in \{0,1\}$. We create preference pairs $(r^+, r^-)$ where $r^+$ is a correct refinement ($\mathbf{V}(r^+) = 1$) and $r^-$ is an incorrect refinement ($\mathbf{V}(r^-) = 0$) for the same verifier input $v_{i,j,k}$. Formally:

$$\mathcal{D}_R = \big\{(r^+, r^-) \mid r^+, r^- \in \{r_{i,j,k,l}\}_{l=1}^{n},\ \mathbf{V}(r^+) = 1,\ \mathbf{V}(r^-) = 0\big\},$$

$$\mathcal{D}_{\mathcal{V}} = \big\{(v^+, v^-) \mid v^+, v^- \in \{v_{i,j,k}\}_{k=1}^{n},\ \hat{v}^+ = \checkmark,\ \hat{v}^- = \times\big\}.$$

For each verifier output $v_{i,j,k}$, we compute its value $\mathbf{V}(v_{i,j,k})$ and binarize it as $\hat{v}_{i,j,k} = \checkmark$ if $\mathbf{V}(v_{i,j,k}) > 0.5$, and 0 otherwise. Preference pairs for the verifier model are created by comparing outputs under the same generator output $g_{i,j}$: A similar process applies to the generator model, where generator outputs $g_{i,j}$ are binarized based on their values $\mathbf{V}(g_{i,j})$, and preference pairs $\mathcal{D}_G$ are created by comparing outputs under the same query $q_i$. This tree-expansion process leads to a sufficiently large (see Sec. 5) and diverse dataset.

*Refinement and Verifier Model Training*   Following the data generation process outlined above and the training setups in Section 3, we first perform SFT on the $G$, $V$, $R$ models with the questions and positive samples in $\mathcal{D}_G$, $\mathcal{D}_{\mathcal{V}}$, and $\mathcal{D}_R$ respectively, updating each model's policy separately. For $G$, this is analogous to standard STaR post-training (Zelikman et al., 2022) with a specialized dataset. Next, on only the SFT updated $V$ and $R$ models, we apply DPO with question and chosen-rejected pairs in $\mathcal{D}_{\mathcal{V}}$ and $\mathcal{D}_R$ respectively. For the Generator, we opt for SFT-only because DPO does not improve performance (primarily because the base LLM - Llama 3.1 8B is already extensively post-trained on preferences as a generator (Grattafiori et al., 2024) and Sec. 5.3). By combining SFT with preference-based updates, we capture both "ideal" behaviors (through correct samples) and "relative" preferences (through correct-vs-incorrect pairs). This allows us to not only bootstrap reasoning based on positive traces but also learn generalizable knowledge about useful reasoning trajectories (Chu et al., 2025). We depict our method in Figure 1 and algorithm in Appendix A.1.

# 5 Experiments

Here, we outline the experimental details, including a description of the model, the benchmarks used for evaluation, and the training pipeline. We then present our main experimental results, followed by an empirical analysis and baseline comparisons, along with ablations.

## 5.1 Experimental Details

**Benchmarks and Models**   We use Llama-3.1-8B-Instruct (Grattafiori et al., 2024), chosen for its open-source nature and balance between competitive baseline performance and size fitting in a limited compute budget. We evaluate MALT and all baselines on three widely-used benchmarks: GSM8K (Cobbe et al., 2021), with 7.47k training examples and 1.32k test questions, focused on diverse grade school math problems. For more challenging mathematical reasoning questions, we use MATH (Hendrycks et al., 2021), with 7.5k training and 5k test problems. MATH has proven to be consistently difficult for smaller language models, with Llama 3.1 8B performing around 49.50% test-accuracy.

Lastly, to extend the scope beyond mathematical tasks, we evaluate on CommonsenseQA (CSQA) (Talmor et al., 2019) with 9.74k training examples and 1.22k dev-set questions. CSQA is a multiple-choice question answering dataset around commonsense reasoning problems and has been used similarly by prior work (Zelikman et al., 2022; 2024; Wei et al., 2023).

**Baselines**   We compare MALT against eight baselines in 2 primary settings, all using Llama-3.1-8B. First, we implement the *inference-only* setting with **(1)** a single-agent (SA) naive setting in which a single model is used as a generator to provide an outupt, **(2)** a multi-agent (MA) setting, where the pre-trained baseline model operates in a sequential way as a generator, verifier, and refinement agent with the same prompts given to MALT post-trained models. Our second setting (*equal training compute STaR baseline*) is with SFT on all three models with the positive synthetic data. We also compare against an equal inference compute multi-agent debate baseline (Du et al., 2024) with 3 agents over 3 rounds.

**MALT Procedure**   For MALT, we generate synthetic data for each benchmark separately using the tree-based approach (Algorithm 1) with a branching factor of $n = 3$, yielding 27 trajectories per question and approximately 2k–6k labeled pairs per model/benchmark. Each final answer is compared against the ground truth in the training set to assign a binary reward, which is then propagated to label the $G$, $V$, and $R$ outputs. During this, each model has a fixed role conditioning prompt template that is also used for baselines and for the post-trained models. We first train each role-specific Llama-3.1-8B-Instruct model on positive labels with LoRA-based SFT and then with DPO to incorporate reasoning preference data for reinforcement learning (discussed in Section 4.3. We use LoRA adapter-based fine-tuning (Hu et al., 2021), reducing the computational load for post-training (see Appendix A.9 for more details). At inference, MALT follows a simple sequential inference pass through the three heterogeneous models reasoning over questions in the test set. Training small models on long sequences of text can lead to instability/hallucinations (Park et al., 2024), and thus MALT and baselines employ a three-vote self-consistency mechanism to mitigate this.

## 5.2 Experimental Results

Our experimental results along with all the baselines are presented in Table 1, along with Ablations in Tables 4 and 5. Our results are averaged over four runs on random subsets of the large test-sets across seeds, and we report the standard deviation for all our core results.

**Baselines comparison**   We present baseline results in Table 1. Baseline single agent scores on MATH, CSQA, and GSM8K are 49.50%, 74.50%, and 84.25%, approximately in line with scores reported in Llama-3.1-8B-Instruct's release (Grattafiori et al., 2024). These scores go up to 52.50%, 75.75%, 86.75% and 53.50%, 79.00%, 87.00% with single model majority voting and multi-agent majority voting respectively. Additionally, after applying STaR (SFT on

| Benchmark | Inference-only | | | | STaR Training | | | | MALT _(w/o MV)_ | MALT |
|---|---|---|---|---|---|---|---|---|---|---|
| | SA | SA+MV | MA | MA+MV | SA | SA+MV | MA | MA+MV | | |
| | | | | | **Test Accuracy (%) over 4 seeds ↑** | | | | | |
| **Llama-3.1-8B-Instruct** | | | | | | | | | | |
| GSM8K | 84.25 ±2.28 | 86.75 ±2.38 | 84.75 ±2.86 | 87.00 ±4.00 | 81.75 ±0.83 | 84.75 ±2.68 | 80.00 ±1.58 | 86.75 ±2.28 | 83.50 ±2.18 | **90.50** ±2.06 |
| CSQA | 74.50 ±3.35 | 75.75 ±5.49 | 77.50 ±5.17 | 79.00 ±2.55 | 76.25 ±4.32 | 78.75 ±4.26 | 75.50 ±2.69 | 76.00 ±1.73 | 77.50 ±1.12 | **81.50** ±2.29 |
| MATH | 49.50 ±2.06 | 52.50 ±2.50 | 51.75 ±3.56 | 53.50 ±2.87 | 52.25 ±1.48 | 54.00 ±2.73 | 52.50 ±3.20 | 53.75 ±2.68 | 52.25 ±1.79 | **57.25** ±1.48 |
| **Qwen-2.5-1.5B-Base** | | | | | | | | | | |
| GSM8K | 61.75 ±4.72 | 65.0 ±3.56 | 60.50 ±2.89 | 62.25 ±2.22 | 63.50 ±1.73 | 64.25 ±2.06 | 62.50 ±4.20 | 64.25 ±2.87 | 71.00 ±3.56 | **74.50** ±2.65 |

Table 1: **Benchmark results.** We compare MALT with baselines on three different benchmarks using Llama-3.1-8B-Instruct, and also evaluate GSM8K performance on Qwen-2.5-1.5B-Base. For baselines, we include different setups such as single agent (SA) and multi-agent (MA), both with and without equal inference compute based majority voting (MV). _MALT outperforms all baselines across both models._

| Method | MATH | CSQA | GSM8K |
|---|---|---|---|
| Inference-only SA | 49.50 | 74.50 | 84.25 |
| Inference-only SA | 49.50 | 74.50 | 84.25 |
| Multi-Agent Debate | 52.00 | 71.25 | 86.75 |
| MALT | **57.25** | **81.50** | **90.50** |

Table 2: **Debate Baseline.** MALT significantly outperforms an equal-inference spend multi-agent debate baseline (3 models debating for 3 rounds) across MATH, CSQA, and GSM8K.

| Method | Accuracy (%) |
|---|---|
| Base Model (Llama 3.1 8B Instruct) | 71.75 ± 2.17 |
| Base Model + MV | 73.75 ± 2.38 |
| Multi-agent + MV | 75.25 ± 2.38 |
| STaR + MV | 76.75 ± 4.60 |
| MALT (Llama 3.1 8B Instruct) | 84.75 ± 3.30 |
| Llama 3.1 70B Instruct | **88.25** ± 2.95 |

Table 3: **Generalization.** MALT generalizes from GSM8K to GSM-Symbolic P1 (GSM-SP1).

positive synthetic CoTs), single-agent baselines achieve around 52.25%, 76.25%, and 81.75% on MATH, CSQA, and GSM8K, while the multi-agent STaR variant remains around 52.50%, 75.50%, and 80.00%. Self consistency results for STaR variants surpass untrained baselines but still underperform improvements obtained with MALT. We also evaluate Qwen-2.5-1.5B-Base (Qwen et al., 2025) for a more diverse analysis. We observe that STaR training does not improve performance significantly beyond inference only baselines. However, DPO training for only the generator achieves a 68.75% accuracy and works much better on non-instruction tuned (base) models. This allows MALT to offer even higher improvements.

**MALT core results** MALT with Llama-3.1-8B-Instruct (Table 1, right) achieves an accuracy of 57.25%, 81.50%, and 90.50% on MATH, CSQA, and GSM8K. Overall, _MALT significantly outperforms all baselines, including all settings with supervised fine-tuned models. Over the untrained model's performance as a generator, MALT achieves relative improvements of_ 15.66%, 9.40%, _and_ 7.42% _on MATH, CSQA, and GSM8K._ This shows the efficacy of our search and attribution based data generation, post-training, and inference pipeline in MALT across benchmarks of varying difficulty. MALT on Qwen-2.5-1.5B-Base, when evaluated using GSM8K, achieves an accuracy of 74.50% and offers a relative improvement of 20.65% over the single agent baseline. We also compare MALT with an equal inference multi-agent debate baseline in Table 2, where it significantly outperforms across all three benchmarks.

**Generalization to more difficult benchmarks** To understand whether MALT on a small open model generalizes to challenging reasoning tasks, we evaluate the Llama-3.1-8B model trained with MALT across a similar setup with increased complexity. We test our GSM8K-based MALT setup on GSM-Symbolic-P1 (Mirzadeh et al., 2024), a complex and much more challenging reasoning benchmark. From results in Table 3, MALT on Llama-3.1-8B achieves an 84.75 ± 3.30% accuracy, outperforming all baselines. Interestingly, MALT also _yields performance comparable to the significantly larger Llama-3.1-70B, which results in_ 88.25 ± 2.95%.

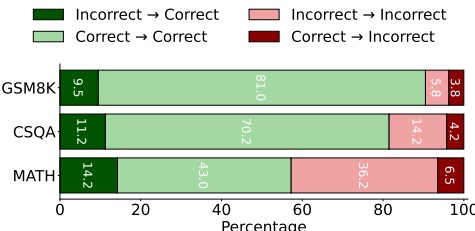

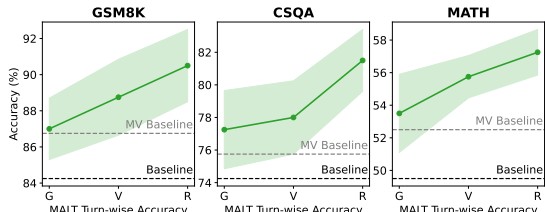

Figure 2: **Self-correction.** MALT consistently increases the number of correct answers by correcting previously incorrect answers at a much higher rate than introducing new mistakes compared to a single-model baseline (MV@3).

Figure 3: **Improvement over turns.** MALT demonstrates consistent improvements at each turn. Test accuracy at each turn from a sequential pass through the post-trained Generator, Verifier, and Refinement models (All with MV@3).

**Self-consistency**   From the Llama 3.1 results in Table 1, MALT shows improved reasoning with majority-voting for self-consistency (Wang et al., 2022), but its performance without MV remained close to that of the multi-agent inference-only setting. Qualitatively, MALT solves questions unsolved by any baseline but suffers occasional hallucinations without voting. Thus, we use a small majority voting factor of 3 and observe that self-consistency reliably yields a higher relative improvement for MALT over baselines. For instance, on MATH, self-consistency resulted in a relative improvement of 9.57%, exceeding the $3 - 6\%$ gains in other baselines. In Figure 2, we measure how often MALT flips an incorrect solution into a correct one versus the opposite. We show that our post-trained generator-verifier-refiner process results in strong self-improvement with a low rate of mistakes introduced.

**Improvement over turns**   We use GPT-4 to rate the correctness of each step produced by MALT (on Llama 3.1-8B-Instruct) on the test sets to analyze its performance at each turn. Figure 3 shows how performance evolves. For e.g., the untrained baseline on MATH improves from 49.50% to 52.50% with majority voting. MALT, over its 3 reasoning and multi-agent deliberation turns, increases performance to 53.50% (turn 1 - Generator), 55.75% (turn 2 - Verifier), and 57.25% (turn 3 - Refiner). We observe similar increases across GSM8K and CSQA. This turn-wise improvement highlights how sequential verification and refinement are both very important for improving reasoning performance. We provide ablations to understand why generate-verify-refine is the most appropriate paradigm in Section 5.3.

**Overall performance**   Across all benchmarks, MALT outperforms both zero-shot and fine-tuned baselines, closing gaps on problems previously unsolved by any baseline. Its multi-agent approach not only achieves higher average scores but also corrects systematic errors. For example, the verifier often successfully locates errors by redoing calculations and providing a precise critique, with the behaviors learnt automatically from synthetic data generated via our search process. Similarly, for CSQA, the verifier implicitly learns to focus on aspects of the problem overlooked by the generator, with examples of reasoning interactions provided in Appendix A.4. Additionally, MALT can be used on any frontier model since it does not rely on the presence of an oracle model for supervision.

## 5.3   Ablation studies

**Untrained model ablations**   We test the usefulness of training each individual agent (Llama MALT setup) by replacing one with an untrained baseline and keeping the other two MALT agents. As seen in Table 4, this degrades performance on all benchmarks. For instance, using an untrained generator yields mean accuracies of 54.25% (MATH), 87.00% (GSM8K), and 78.75% (CSQA), notably lower than our trained system, showing how *all* agents benefit from the MALT pipeline.

**Importance of generate-verify-refine**   Next, we measure the impact of ablating one role from our three-agent pipeline (Llama MALT setup) for simpler two-agent systems. We compare (i) *generate-refine* ($G + R$, skipping the verifier) and (ii) *generate-verify* ($G + V$, skipping the refiner). As shown in Table 5, both setups underperform the full pipeline:

| MALT Post-training | | | GSM8K | CSQA | MATH |
|---|---|---|---|---|---|
| $G$ | $V$ | $R$ | | | |
| ✗ | ✓ | ✓ | 87.00 | 78.75 | 54.25 |
| ✓ | ✗ | ✓ | 85.75 | 76.75 | 54.50 |
| ✓ | ✓ | ✗ | 86.25 | 75.50 | 55.25 |
| ✓ | ✓ | ✓ | **90.50** | **81.50** | **57.25** |

Table 4: **Ablations with untrained models (MV@3).** Combining untrained agents with trained ones shows that all LLMs perform best when cooperating with MALT agents.

| Configuration | GSM8K | CSQA | MATH |
|---|---|---|---|
| $G$ only | 84.75 | 78.75 | 54.00 |
| $G + V$ | 88.75 | 78.00 | 55.75 |
| $G + R$ | 84.75 | 76.25 | 54.75 |
| $G + V + R$ (Ours) | **90.50** | **81.50** | **57.25** |

Table 5: **Performance of ablated multi-agent setups (MV@3).** Our experiments show that all agents in the MALT pipeline are necessary to achieve the best results.

$G + R$ yields mean accuracies of 54.75% (MATH), 84.75% (GSM8K), and 76.25% (CSQA), while $G + V$ yields 55.75% (MATH), 88.75% (GSM8K), and 78.00% (CSQA), showing that our specific multi-agent setup yields stronger results by spending inference for improved sequential reasoning. Importantly, we note that our three-model pipeline does not rigidly fix MALT's usability. It is, instead, a modular and flexible entry-point for multi-agent training. MALT subsumes the utility of a single model learning from more data, while providing an opportunity to better use specialized models for improved reasoning. With our search and credit assignment strategy, MALT reduces the risk of reinforcing a model's existing biases, providing diverse training signal for system-level improvement that allows for generalization across difficult problems.

**Effectiveness of DPO over only SFT**   As shown in Table 1, DPO improves performance beyond SFT alone (STaR) by using negative data—as observed in (Putta et al., 2024; Setlur et al., 2024). In particular, purely positive "rationales" can introduce spurious correlations and degrade SFT performance, which does indeed occur in our empirical and qualitative results; the contrastive training approach that DPO provides instead helps the model identify high-advantage reasoning steps to improve with higher sample-efficiency (Rafailov et al., 2024). For reasoning problems, SFT tends to memorize the data and rules, which is useful to bootstrap reasoning to a certain extent (Zelikman et al., 2022). However, our results indicate that this could degrade performance sometimes, and preference optimization methods (see Appendix A.6.2 for a theoretical analysis and A.9 for possible issues) exhibit better performance at learning generalizable knowledge for reasoning steps (Chu et al., 2025).

## 6   Discussion and Conclusion

We presented MALT, a novel post-training strategy dividing CoT reasoning among three specialized LLMs to tackle complex problems. MALT bridges the gap between prior multi-agent inference methods and fully-trained multi-agent systems by generating role-specific data using a tree-based sampling and credit assignment mechanism. Crucially, MALT utilizes the negative synthetic data to identify and correct flawed reasoning steps with LLM post-training, improving role-specific reasoning capabilities. Unlike standard single-LLM setups, our design closely mirrors how humans solve complex tasks or even use LLMs—attempting a solution, critiquing errors, and finally refining the result. While we were unable to expand experiments to significantly larger models due to compute limitations, we have evaluated against a comprehensive set of challenging datasets and benchmarks. Our generator-verifier-refinement setup focuses on improving meta-strategies such as self-correction or chaining inference steps, but it can be extended to automatically learnt roles obtained via search. We discuss this, along with future work such as repeated rounds of refinement, online RL settings, and scaling the branching factor in Section A.8.

**Safety**   Our approach can be used not just to enhance the reasoning capabilities of LLM systems, but also address crucial open problems in the safety of multi-agent systems. Importantly, MALT-trained systems of trusted small models could attain better task performance while retaining high degrees of trust, producing more powerful overseers within the AI control setting (Greenblatt et al., 2024). Another prominent application of our approach would be to train verifiers as safety critics. This could scale up the settings such as OpenAI CriticGPT (McAleese et al., 2024) to any number of models, resulting in more powerful safety critics and allowing for the legibility of solutions to be improved.

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

# A  Appendix

## A.1  Algorithm

MALT orchestrates a three-agent system—comprising a Generator for initial answers, a Verifier for critiques, and a Refiner that integrates these steps into a final response. During training, we expand a multi-agent search tree for each question, labeling correct and incorrect branches via a value iteration based credit assignment mechanism. This generates role-specific preference pairs for post-training each agent via supervised fine-tuning and preference optimization. Algorithm 1 provides a complete description of the data collection and training pipeline for MALT.

---

**Algorithm 1** Multi-Agent LLM Training and Synthetic Data Generation (MALT)

---

**Require:** Initial Dataset $\mathcal{D}$, Models $G$, $V$, $R$, Branching factor $n$
    **GOAL:** Trained models $G'$, $V'$, $R'$
1: Initialize datasets $\mathcal{S}_G$, $\mathcal{S}_V$, $\mathcal{S}_R$ as empty sets
2: **for** $q \in \mathcal{D}$ **do**
3:     $A_G \leftarrow \{g_j = G(q)\}_{j=1}^n$                                            ▷ Generate $n$ outputs from $G$
4:     **for** each $g_j \in A_G$ **do**
5:         $A_V^{g_j} \leftarrow \{v_{j,k} = V(q, g_j)\}_{k=1}^n$                            ▷ Generate $n$ outputs from $V$
6:         **for** each $v_{j,k} \in A_V^{g_j}$ **do**
7:             $A_R^{g_j,v_{j,k}} \leftarrow \{r_{j,k,l} = R(q, g_j, v_{j,k})\}_{l=1}^n$           ▷ Generate $n$ outputs from $R$
8:             **for** each $r_{j,k,l} \in A_R^{g_j,v_{j,k}}$ **do**
9:                 Compute $\mathcal{V}(r_{j,k,l}) = \mathcal{R}(r_{j,k,l}, a^{\text{GT}})$         ▷ Reward for $R$ output
10:                 Add $(q, g_j, v_{j,k}, r_{j,k,l}, \mathcal{V}(r_{j,k,l}))$ to $\mathcal{S}_R$
11:             **end for**
12:             Compute $\mathcal{V}(v_{j,k}) = \frac{1}{n} \sum_{l=1}^n \mathcal{V}(r_{j,k,l})$          ▷ Value $\mathcal{V}$ for $V$ output
13:             Binarize $\hat{v}_{j,k} = \mathbb{I}[\mathcal{V}(v_{j,k}) > 0.5]$
14:             Add $(q, g_j, v_{j,k}, \hat{v}_{j,k})$ to $\mathcal{S}_V$
15:         **end for**
16:         Compute $\mathcal{V}(g_j) = \frac{1}{n} \sum_{k=1}^n \mathcal{V}(v_{j,k})$         ▷ Value $\mathcal{V}$ for $G$ output
17:         Binarize $\hat{g}_j = \mathbb{I}[\mathcal{V}(g_j) > 0.5]$
18:         Add $(q, g_j, \hat{g}_j)$ to $\mathcal{S}_G$
19:     **end for**
20: **end for**
21: **Training the Models**
22: $G' \leftarrow \text{SFT}(G, \mathcal{S}_G)$                              ▷ Fine-tune $G$ with supervised data
23: $V_{\text{SFT}} \leftarrow \text{SFT}\left(V, \left\{(q, g_j, v_{j,k}) \mid (q, g_j, v_{j,k}, \hat{v}_{j,k}) \in \mathcal{S}_V, \hat{v}_{j,k} = 1\right\}\right)$   ▷ Fine-tune $V$ on positive samples
24: $V' \leftarrow \text{DPO}(V_{\text{SFT}}, \mathcal{S}_V)$                       ▷ Train $V$ with DPO using preferences
25: $R_{\text{SFT}} \leftarrow \text{SFT}\left(R, \left\{(q, g_j, v_{j,k}, r_{j,k,l}) \mid \mathcal{V}(r_{j,k,l}) = 1\right\}\right)$   ▷ Fine-tune $R$ on positive samples
26: $R' \leftarrow \text{DPO}(R_{\text{SFT}}, \mathcal{S}_R)$                    ▷ Train $R$ with DPO using preferences
27: **return** $G'$, $V'$, $R'$

---

## A.2  Experiments with GSM-Symbolic

GSM-Symbolic (Mirzadeh et al., 2024) is a synthetic extension of GSM8K that uses symbolic templates to produce varied instances (e.g., altering names, numerical values, or adding clauses). The "P1" variant adds an extra clause per question, making the problems more challenging and exposing whether a model relies on shallow memorization or can genuinely handle additional reasoning steps. Testing MALT (Llama 3.1 8B Instruct model trained on

the GSM8K training set) on GSM-Symbolic P1 allows us to understand whether the multi-agent setup allows for robust multi-step reasoning beyond the original data distribution. In such cases, even though the base model is much smaller compared to Llama 3.1 70B, it can find mistakes in subsequent iterations. Given that GSM-Symbolic tests for reasoning and penalizes memorization, we see MALT achieving a performance comparable to Llama-3.1-70B Instruct.

| Method | Accuracy (%) |
|---|---|
| Base Model (Llama 3.1 8B Instruct) | $71.75 \pm 2.17$ |
| Base Model + MV | $73.75 \pm 2.38$ |
| Multi-agent + MV | $75.25 \pm 2.38$ |
| STaR + MV | $76.75 \pm 4.60$ |
| MALT (Llama 3.1 8B Instruct) | $84.75 \pm 3.30$ |
| Llama 3.1 70B Instruct | $\mathbf{88.25} \pm 2.95$ |

Table 6: **Comparison with a larger model.** MALT on Llama 3.1 8B Instruct outperforms a significantly larger model (Llama 3.1 70B Instruct) on GSM-Symbolic P1, demonstrating an important jump in reasoning performance on challenging benchmarks. Results are over 4 seeds for a random subset of the GSM-Symbolic P1.

### A.3   Multi-Agent Debate Baseline

Based on (Du et al., 2024), we provide results for a multi-agent debate baseline. In order to ensure a fair comparison between MALT and the multi-agent debate baseline in terms of inference spend, we used a baseline with 3 agents debating over 3 rounds. We present our results in Table 7 and note that MALT clearly outperforms an equal inference compute multi-agent debate baseline across all three benchmarks.

| Benchmarks | MATH | CSQA | GSM8K |
|---|---|---|---|
| Base Model | 49.50 | 74.50 | 84.25 |
| Multi-Agent Debate | 52.00 | 71.25 | 86.75 |
| MALT | **57.25** | **81.50** | **90.50** |

Table 7: **Debate.** MALT significantly outperforms an equal-inference spend multi-agent debate baseline.

### A.4   Qualitative Examples

In this section, we present three key qualitative examples: a synthetic training example for SFT and DPO, a qualitative performance comparison between an untrained and a trained verifier, and a complete MALT setup demonstrating a backtracking and improvement reasoning cycle that generates a correct answer.

---

**GSM8K Synthetic Dataset Sample for the Refinement Model (GT Answer = 25)**

**System prompt**
*You are an expert problem solver that refines math solutions based on feedback.*

**Input**
*Question: Marco loves trading cards. He has a total of 500 cards on hand, and a fourth of them are duplicates. Marco decides to trade one-fifth of these duplicates for completely new cards to his friend Josh. How many new cards did Marco get?*
*Initial Answer: I'd be happy to help you solve this problem step by step.*
Step 1: Identify the key information given in the question
* Marco has a total of 500 cards. * A fourth of them are duplicates.

---

Step 2: Determine what the question is asking us to find
* The question is asking us to find how many new cards Marco got from trading one-fifth of his duplicates with Josh.
Step 3: Plan our approach to solving the problem
* First, we need to find the number of duplicates Marco has. Since a fourth of his cards are duplicates, we can find the number of duplicates by multiplying 500 by 1/4. * Then, we need to find one-fifth of the number of duplicates to find out how many cards Marco traded with Josh. * Finally, we'll subtract the number of cards traded from the total number of duplicates to find the remaining duplicates.
Step 4: Execute our plan
* Number of duplicates = 500 x 1/4 = 125 * One-fifth of the number of duplicates = 125 x 1/5 = 25 * Marco traded 25 cards with Josh.
Step 5: State our conclusion and double-check our answer
* Marco got 25 new cards from trading with Josh.
**Final Answer: 25**

**Rejected Output**
*Based on the initial answer and the verification feedback, I refine the solution as follows:*
Step 1: Identify the key information given in the question
* Marco has a total of 500 cards. * A fourth of them are duplicates.
Step 2: Determine what the question is asking us to find
* The question is asking us to find how many new cards Marco got from trading one-fifth of his duplicates with Josh.
Step 3: Plan our approach to solving the problem
* First, we need to find the number of duplicates Marco has. Since a fourth of his cards are duplicates, we can find the number of duplicates by multiplying 500 by 1/4. * Then, we need to find one-fifth of the number of duplicates to find out how many cards Marco traded with Josh. * Finally, we'll subtract the number of cards traded from the total number of duplicates to find the remaining duplicates.
Step 4: Execute our plan
* Number of duplicates = 500 x 1/4 = 125 * One-fifth of the number of duplicates = 125 x 1/5 = 25 * Marco traded 25 cards with Josh. Since he traded one-fifth of his duplicates, we can find the total number of duplicates he has left by adding the number of cards traded back to the number of duplicates: 125 + 25 = 150 * To find the number of new cards Marco got, we subtract the number of duplicates he traded from the total number of duplicates: 150 - 25 = 125
Step 5: State our conclusion and double-check our answer
* Marco got 125 new cards from trading with Josh.
**Final Answer: Final Answer: 125**

**Accepted Output**
Refined Solution:
Step 1: Identify the key information given in the question
* Marco has a total of 500 cards. * A fourth of them are duplicates. * Marco trades one-fifth of his duplicates with Josh.
Step 2: Determine what the question is asking us to find
* The question is asking us to find how many new cards Marco got from trading one-fifth of his duplicates with Josh.
Step 3: Plan our approach to solving the problem
* First, we need to find the number of duplicates Marco has. Since a fourth of his cards are duplicates, we can find the number of duplicates by multiplying 500 by 1/4. * Then, we need to find one-fifth of the number of duplicates to find out how many cards Marco traded with Josh. * Finally, we'll subtract the number of cards traded from the total number of duplicates to find the remaining duplicates.
Step 4: Execute our plan

* Number of duplicates = 500 x 1/4 = 125 * One-fifth of the number of duplicates = 125 x 1/5 = 25 * Marco traded 25 cards with Josh. * Remaining duplicates = 125 - 25 = 100
Step 5: State our conclusion and double-check our answer
* Marco got 25 new cards from trading with Josh.
**Final Answer: 25**

---

**Qualitative Example of an Untrained VS Trained Verifier**

**Question**
*Billy buys a 12-pack of soda from the store. If he has twice as many brothers as sisters, and he has 2 sisters, how many sodas can he give to each of his siblings if he wants to give out the entire 12-pack while giving each the same number?*

**Initial Answer**
*I'd be happy to help you solve this problem step by step.*
**Step 1: Identify the key information given in the question.**
* Billy buys a 12-pack of soda. * He has twice as many brothers as sisters. * He has 2 sisters.
**Step 2: Determine what the question is asking us to find.**
The question is asking us to find out how many sodas Billy can give to each of his siblings if he wants to give out the entire 12-pack while giving each the same number.
**Step 3: Plan our approach to solving the problem.**
Let's let x be the number of siblings (brothers and sisters). We know that Billy has 2 sisters, so the number of brothers is 2 times the number of sisters, which is 2x. The total number of siblings is the sum of the number of sisters and brothers, which is x + 2x = 3x. We are given that Billy has a 12-pack of soda, so we can set up the equation:
12 = x + 2x
**Step 4: Execute our plan, showing all work clearly.**
To solve for x, we can combine like terms:
12 = 3x
Divide both sides by 3:
4 = x
So, Billy has 4 siblings in total (2 sisters and 2 brothers).
**Step 5: State our conclusion and double-check our answer.**
Since Billy has 12 sodas and 4 siblings, he can give:
12 ÷ 4 = 3
sodas to each of his siblings.
**Final Answer: 3**

**Untrained Verifier**
*As a verifier, I will critically evaluate the initial answer provided above.*
The initial answer states that Billy can give 3 sodas to each of his siblings. Let's verify the answer by performing the calculations.
**Verification:**
1. We are given that Billy has 12 sodas and 4 siblings (2 sisters and 2 brothers). 2. To find out how many sodas each sibling can get, we can divide the total number of sodas (12) by the number of siblings (4):
12 ÷ 4 = 3
The initial answer is correct! Billy can indeed give 3 sodas to each of his siblings.
**Feedback:**
The initial answer is mathematically correct. The calculations are accurate, and the solution is well-explained. The answer is also logically consistent with the given information. Well done!

However, it's always a good idea to double-check the calculations to ensure accuracy. In this case, the verification step confirms that the initial answer is correct.

### Trained Verifier

*As a verifier, I will carefully evaluate the initial answer provided above.*

The initial answer states that Billy can give 3 sodas to each of his siblings. Let's double-check the calculations to ensure the answer is mathematically correct.

The initial answer claims that Billy has 4 siblings (2 sisters and 2 brothers). However, the question states that Billy has twice as many brothers as sisters, and he has 2 sisters. This implies that the number of brothers is 2 times the number of sisters, which is 2 x 2 = 4 brothers. Therefore, Billy actually has 2 sisters and 4 brothers, making a total of 6 siblings.

The initial answer also claims that Billy can give 12 sodas to 4 siblings, resulting in 3 sodas per sibling. However, dividing 12 sodas among 6 siblings would actually give 2 sodas per sibling, not 3.

**Conclusion:**

The initial answer contains a calculation error. The correct number of siblings is 6, not 4. Therefore, Billy can give 2 sodas to each of his siblings, not 3.

As a verifier, I recommend that the initial answer be corrected to reflect the correct calculation. The correct answer is 2 sodas per sibling.

---

### A Complete Reasoning and Improvement Sequence Example with MALT on CommonsenseQA

### Question

He looked at the field of pumps, all slowly churning oil out of the what?

### Choices

A: manual
B: street
C: restaurant
D: ground
E: service station

### Initial Answer (Generator)

Reasoning steps: 1. The context of the question is about pumps churning oil, which suggests a location where oil is being dispensed or extracted. 2. A "manual" (A) does not typically relate to a location where oil is churning. 3. A "street" (B) is a general location, but it's not specific to oil churning. 4. A "restaurant" (C) is not associated with oil churning. 5. The "ground" (D) could be a possible location, but it's not the most specific or common location for oil churning. 6. A "service station" (E) is a common location where oil is churning, as it's where cars are serviced and oil is dispensed.

**Final Answer: E**

### Verifier

The initial answer is E: service station.

This answer seems to be the most logical choice given the context of the question. The phrase "field of pumps" suggests a location where multiple pumps are present, and a service station is a common place where multiple pumps are used to dispense oil and other fuels.

However, the question asks what the pumps are churning oil out of, not where the pumps are located. This changes the interpretation of the question.
Considering the phrase "churning oil out of," it implies that the pumps are extracting or pumping oil from a source. In this case, the most likely source of oil would be the ground, as oil is often extracted from the ground.
**Therefore, the correct answer is D: ground.**

**Refined Answer**
Reasoning steps: 1. The context of the question is about pumps churning oil, which suggests a location where oil is being extracted or dispensed. 2. The phrase "churning oil out of" implies that the pumps are extracting or pumping oil from a source. 3. The most common source of oil is the ground, as it is often extracted from the earth. 4. Therefore, the pumps are churning oil out of the ground.
**Final Answer: D**

**Ground Truth (not provided to any model)**
D

## A.5 Detailed Discussion

We now discuss key design choices. Simplicity with theoretical grounding—particularly the assurance of a trained policy with a monotonically increasing value—was a primary consideration. We opted for DPO over PPO-based methods due to our focus on a straight-forward offline data generation process, treating our approach as an independent learning problem in a centralized setting (Lerer et al., 2020) with a single iteration (the key difference being that our agent policies post-training differ). In this setting, DPO is more stable than PPO and requires less overhead. While PPO could use the value computed at each branch as a reward for post-training nodes (a promising future direction), it introduces significant computational complexity. Moving from offline to online RL with additional computational overhead might indeed improve performance.

Our value iteration method, when binarized, resembles global majority-based pooling: for a given node and branch, the binary reward of the leaf nodes in the subtree determines the usefulness of the branch, analogous to binarizing values propagated through the tree. In contrast, local pooling computes the binary value of a branch based only on the majority outcomes of its direct children, propagating this process to the leaf nodes. We also leave the choice between MCTS and an expansive tree-based sampling strategy as an open problem. Given our limited tree depth, tree-based sampling proved efficient, supported synthetic data generation with an exponential branching factor, and produces explainable outputs. Our dataset is collected offline, and individual models are trained on this synthetic data. While this approach works empirically, handling any new, out-of-distribution data would require iterative rollout and post-training methods.

Based on our empirical results and the modularity of our algorithmic approach, it is highly plausible that our method will scale to larger models and scenarios with many agents, thus laying the foundations for new state-of-the-art AI agents based on systems of cooperative frontier models. Overall, our multi-agent system is currently composed of a sequence of agents that start out with the same parameters and different prompts. MALT performs joint training to transform this into a heterogeneous agent setting, where agents with different parameters operate cooperatively. Exploring other multi-agent settings is an important direction for subsequent work.

### A.6 Theoretical Justification for MALT

#### A.6.1 Credit Assignment Strategy

Here, we provide a theoretical justification for why our framework, when updating the agent policies based on binarized pooled rewards with a threshold at $\theta = 0.5$, leads to policy improvements. We formalize MALT as a three-step MDP, define the pooling operation through value iteration, and demonstrate how off-policy updates increase the expected reward.

The reasoning process in MALT is modeled as a three-step MDP over a set of $M$ questions $\{q_i\}_{i=1}^{M}$ drawn from a distribution $Q$. For each question $q_i$, the process begins at the initial state $s_0 = q_i$, where an initial answer $g_{i,j}$ is sampled from the generator policy $\pi_G(\cdot \mid s_0)$ for $j = 1, \ldots, n$. The state then transitions to $s_1 = (q_i, g_{i,j})$. At this second state, a critique $v_{i,j,k}$ is sampled from the verifier policy $\pi_V(\cdot \mid s_1)$ for $k = 1, \ldots, n$, leading to the state $s_2 = (q_i, g_{i,j}, v_{i,j,k})$. Finally, at this state, a refined answer $r_{i,j,k,l}$ is sampled from the refiner policy $\pi_R(\cdot \mid s_2)$ for $l = 1, \ldots, n$, and a reward $R(s_2, r_{i,j,k,l})$ is assigned: 1 if $r_{i,j,k,l}$ is correct, 0 otherwise. The joint policy is defined as $\pi = (\pi_G, \pi_V, \pi_R)$, and the objective can now be expressed as:

$$J(\pi) \;=\; \mathbb{E}_{q \sim Q}\Big[\mathbb{E}_{g \sim \pi_G(\cdot|s_0)} \, \mathbb{E}_{v \sim \pi_V(\cdot|s_1)} \, \mathbb{E}_{r \sim \pi_R(\cdot|s_2)} \big[R(s_2, r)\big]\Big].$$

Reasoning trajectories are collected offline under an initial policy $\pi^{(0)} = (\pi_G^{(0)}, \pi_V^{(0)}, \pi_R^{(0)})$, yielding $M \cdot n^3$ total samples. Through this tree-sampling method, values propagate backward using value iteration: Leaf nodes have $\mathcal{V}(r_{i,j,k,l}) = R(r_{i,j,k,l}) \in \{0,1\}$, verifier nodes compute $\mathcal{V}(v_{i,j,k}) = \frac{1}{n}\sum_{l=1}^{n}\mathcal{V}(r_{i,j,k,l})$, and generator nodes estimate $\mathcal{V}(g_{i,j}) = \frac{1}{n^2}\sum_{k=1}^{n}\sum_{l=1}^{n}\mathcal{V}(r_{i,j,k,l})$. The true value $\mathcal{V}^*(v) = \mathbb{E}_{\pi^{(0)}}[R \mid v]$ is approximated by these Monte Carlo estimates.

We note that our analysis rests on the coverage assumption, where for any relevant action (e.g., $g$ with $\mathcal{V}^*(g) > \mathbb{E}[\mathcal{V}^*(g)]$) over $\pi_G^{(0)}$, the initial policy satisfies $\pi_G^{(0)}(g \mid q) \geq \alpha > 0$, with analogous conditions for $\pi_V^{(0)}$ and $\pi_R^{(0)}$.

Independence holds across levels: refinements $r_{i,j,k,l}$ are i.i.d. given $(g_{i,j}, v_{i,j,k})$, critiques $v_{i,j,k}$ are conditionally independent given $g_{i,j}$, answers $g_{i,j}$ are i.i.d. given $q_i$, and questions $q_i$ are from $Q$. Moreover, our exponential branching factor $n$ allows for sufficient sampling.

This allows us to show that our value estimates are within a certain $\epsilon$ bound of their true values with high probability. For a node $v$ with $m$ downstream refinements (e.g., $m = n^2$ for $g_{i,j}$), the value estimate $\mathcal{V}(v) = \frac{1}{m}\sum_{l=1}^{m} R_l$, where each $R_l \sim \text{Bernoulli}(\mathcal{V}^*(v))$ provides an unbiased estimator of $\mathcal{V}^*(v)$. Hoeffding's inequality bounds the estimation error:

$$P\big(|\mathcal{V}(v) - \mathcal{V}^*(v)| \geq \epsilon\big) \leq 2\exp(-2m\epsilon^2).$$

For generator nodes ($m = n^2$), this becomes:

$$P\big(|\mathcal{V}(g_{i,j}) - \mathcal{V}^*(g_{i,j})| \geq \epsilon\big) \leq 2\exp(-2n^2\epsilon^2).$$

Thus, applying a union bound over all $Mn$ generator nodes, we find that with probability at least $1 - \delta$, the estimation error for any given generator node for all questions satisfies

$$|\mathcal{V}(g_{i,j}) - \mathcal{V}^*(g_{i,j})| \leq \epsilon, \quad \text{where} \quad \epsilon = \sqrt{\frac{\ln\left(\frac{2Mn}{\delta}\right)}{2n^2}}.$$

MALT binarizes node values using a threshold of 0.5: $\hat{\mathcal{V}}(v) = 1$ if $\mathcal{V}(v) > 0.5$, and 0 otherwise. This is analogous to majority-voting, where $\mathcal{V}(v) > 0.5$ indicates that most refinements are correct, aligns with the Bernoulli decision boundary for binary rewards,

and balances misclassification costs for conservative updates. The updated policy $\pi(1)$ shifts probability mass toward these "high-value" nodes using Supervised Finetuning (SFT) and Direct Preference Optimization (DPO). For the refiner, SFT is applied for nodes where $\hat{\mathcal{V}}(r_{i,j,k,l}) = 1$ followed by DPO using preference pairs $(r^+, r^-)$.

Similarly for the verifier, SFT is followed by DPO on its credit assigned preference pairs. For the generator, SFT is performed using answers where $\hat{\mathcal{V}}(g_{i,j}) = 1$. These updates improve the joint policy by prioritizing actions that yield higher expected rewards under $\pi(0)$. In Section A.6.2, we discuss why policy optimizing our DPO objective based on data collected offline under $\pi(0)$ is identical to the optimal RL policy.

Finally, we provide an intuitive explanation of our threshold $\theta$ used for credit assignment. In iterative settings, $\theta$ should be an adaptive factor increasing from 0.5 to 1. However, in our offline setting, 0.5 is a balanced threshold to use for the following reasons:

- Lower Thresholds ($\theta < 0.5$): This allows for greater sample-efficiency as more branches labeled as correct are used as part of training. However, it might introduce noise into the training process with samples that have low values being chosen as correct.

- Higher Thresholds ($\theta > 0.5$): This would allow for a focus on actions leading to higher-value nodes, reducing variance. However, having $\theta$ too high would reduce sample efficiency.

Using $\theta = 0.5$ provides a balance suitable for a single iteration based on an offline generated dataset. By formalizing our value iteration approach and policy updates, we have shown how MALT increases the probability of selecting outputs leading to higher expected return from the system, and thus increases overall multi-agent performance.

### A.6.2 Policy optimizing the DPO objective is identical to Optimal RL Policy

To support our claims in Appendix A.6.1, we leverage Theorem 1 from (Putta et al., 2024) and Theorem 6.1 from (Setlur et al., 2024), adjusted for our binarization setting:

**Theorem.** Consider a policy $\pi$ that optimizes our objective over trajectories generated by a reference policy $\pi_{\text{ref}}$. At each node (state) $h_t$, preferences between actions during DPO are generated according to:

$$p(a_t^w \succ a_t^l \mid h_t) \propto \sigma\left(\hat{Q}(h_t, a_t^w) - \hat{Q}(h_t, a_t^l)\right), \tag{1}$$

where:

- $a_t^w$ and $a_t^l$ are two win/loss actions at node $h_t$,

- $\hat{Q}(h_t, a) \in \{0, 1\}$ is the binarized value function, representing the expected reward of action $a$ at state $h_t$,

Then, the policy that optimizes the Direct Preference Optimization (DPO) objective is identical to the optimal RL policy:

$$\pi^*(a \mid h_t) \propto \pi_{\text{ref}}(a \mid h_t) \exp\left(\frac{\hat{Q}(h_t, a)}{\beta}\right), \tag{2}$$

where $\beta$ is the DPO hyperparameter.

The proof for *Theorem 1* in (Putta et al., 2024) shows that the policy $\pi^*$ approximates the optimal RL policy. That is, we can approximate the optimal RL policy if we generate preferences under the optimal value function (or an approximation thereof, i.e. our binarized version as shown below).

In our setting, since $\hat{Q}(h_t, a) \in \{0, 1\}$, the exponential term simplifies to:

- $\exp\left(\frac{1}{\beta}\right)$ when $\hat{Q}(h_t, a) = 1$,

- $1$ when $\hat{Q}(h_t, a) = 0$.

Therefore, the optimized policy becomes:

$$\pi^*(a \mid h_t) \propto \begin{cases} \pi_{\text{ref}}(a \mid h_t) \exp\left(\frac{1}{\beta}\right), & \text{if } \hat{Q}(h_t, a) = 1, \\ \pi_{\text{ref}}(a \mid h_t), & \text{if } \hat{Q}(h_t, a) = 0. \end{cases} \tag{3}$$

This means that the policy $\pi^*$ increases the probability of selecting actions with $\hat{Q}(h_t, a) = 1$ by a constant factor relative to the reference policy $\pi_{\text{ref}}$. By optimizing the DPO objective with these binarized preferences, we ensure that the policy increasingly favors actions leading to higher expected rewards, aligning with our credit assignment strategy described in Appendix A.6.1. This supports our claim of (approximate) monotonic improvement, as the policy updates move us closer to the optimal policy by consistently selecting actions associated with higher binarized values.

## A.7 Additional Related Work

**Inference Time Compute:** Strategic use of inference-time compute can also boost accuracy. (Brown et al., 2024) demonstrates that repeated sampling from a single model improves coverage, while (Snell et al., 2024) shows that iterative refinement can outperform naive scaling in certain tasks. Meanwhile, supervised fine-tuning (SFT) remains the backbone of LLM adaptation (Devlin et al., 2019; Howard & Ruder, 2018), but can demand large volumes of human-labeled data. Quasi-supervised methods (Yang et al., 2024) address data scarcity by providing supervision for intermediate steps. Finally, (Zelikman et al., 2022) and (Xiang et al., 2025) underscore the effectiveness of structured chain-of-thought prompting for complex tasks, while (Cobbe et al., 2021) confirm that an explicit verifier stage reduces errors on GSM8K. Our work unifies the aforementioned techniques to create a multi-agent pipeline that orchestrates a generator, verifier, and refinement model, leveraging search, synthetic data generation, and preference-based training to enable robust multi-step reasoning.

**Multi-Turn RL Strategies:** Recent work on multi-turn RL (Zhou et al., 2024) focuses on live model interactions over long horizons. MALT instead presents an offline training scheme, where models are post-trained on a preference dataset. Works such as (Shridhar et al., 2023) focus on allowing base LLMs to improve their answers with refinement strategies, and (Qu et al., 2024) further extend such work by allowing for repeated self-improvement turns with the same model. In contrast, MALT focuses on developing a framework where multiple models can reason together to improve overall system performance, and is focused on post-training each model in the system for its specialized role.

## A.8 Future Directions

### A.8.1 Improving Multi-Agent Systems

Our findings showcase the potential of multi-agent LLM systems optimized with fine-tuning and collaborative inference techniques. There are several future directions from this line of work: Using PPO (Schulman et al., 2017) and the exact value propogated backward for each trajectory to update model weights, possibly in an online RL fashion, might produce strong results with additional computational overhead (Ivison et al., 2024). Moreover, we provide several levers around the number of models (where the three model setup can be used iteratively), controlling the branching factor for data generation, examining the effect of majority voting on more samples, changing the attribution threshold, or treating the attribution threshold as an adaptive parameter when iteratively training and rolling out from the multi-agent system (see Appendix A.9). Moreover, prompt-tuning strategies and different roles can be considered or distillation techniques. We note that these are all specific and interesting directions. However, they lie beyond the scope of this paper, where our

goal is to introduce a new multi-agent post-training methodology and demonstrate strong empirical performance.

As for expanding beyond the Generate-Verify-Refine paradigm, it is possible for an orchestrator (human or automated) to either specify different roles or the number of rounds of interaction required. In case repeated interactions are required, the search tree can just be expanded in terms of its depth, and instead of an exponential production of steps, an MCTS process can be used to more efficiently prune our tree to collect synthetic data at each step. Roles can also be decided by a search process, where several proposed roles collaborate and are pruned until a good combination is found for post-training. However, these steps remain beyond the scope of our work, where our goal was to introduce a multi-agent post-training framework that can be conveniently expanded in the near future.

### A.8.2 *Relation to DeepSeek R1 and LLM Mid-training*

With the concurrent release of (DeepSeek-AI et al., 2025, DeepSeek R1) and similar reasoning models, an open gap remains: *How do we generate cold-start data to enable better exploration of meta-strategies with reinforcement learning based training of LLMs?* MALT is not a competing, but a complementary framework to such advances in single-model reasoning. Importantly, it can be used in two specific ways: MALT can provide search trajectories that could be distilled into reasoning paths that can be used for instruction tuning so as to prime the LLM for reinforcement learning (i.e. providing the LLM a baseline level of exploration/self correction/self refinement capabilities so that RL can be significantly more efficient). An alternative approach is to use PPO style online RL methods for multi-agent training with the same credit assignment strategy we have described, instead of collecting offline preferences. This is a viable direction in the presence of more computational resources for training and inference.

### A.9 Additional Information

For SFT, we used LoRA with a learning rate multiplier of 0.1 and a batch size of 8 to avoid overfitting. For preference optimization, we used Direct Preference Optimization (DPO) with a preference tuning learning rate multiplier to 0.1, training beta parameter of 0.2, and adapter weight configured to 0.2. We varied the number of epochs between 1 to 10 based on the size of the synthetic dataset for each model and leave a deeper exploration of hyperparameter configurations that could require a significant amount of compute to future work. SFT training was often until convergence. DPO training did not necessarily converge by the end of all iterations. For the Generator, we find that SFT+DPO actually *lowers* performance (for e.g. 52.25% on MATH with SFT and 51.25% with SFT+DPO)—likely because Llama 3.1 8B-Instruct already underwent post-training with DPO on a very similar generator data distribution for benchmarks (Grattafiori et al., 2024), making DPO on a similar distribution prone to overfitting, consistent with observations in (Setlur et al., 2024).

We keep our prompts the same for every baseline and trained model on a given benchmark. Our prompts use CoT and zero-shot prompting. We use a temperature of 0.3 for Llama 3.1 8B Instruct since it was qualitatively good enough to prevent hallucinations and still led to diverse enough samples. MALT requires the presence of an initial training set containing question-answer pairs, which led to the use of MATH, CSQA, and GSM8K.

During inference for the data collection strategy, using an exponential branching factor does not add significant compute overhead because inference calls can be parallelized when sampling from a model with the exact same input. Moreover, during training, we obtain a fixed dataset upon which LoRA fine-tuning can be conducted. LoRA adapters ensure that the model weights themselves aren't duplicated, thus requiring only minimal additional memory for the adapters themselves while the base models remain the same. For our ablation with only the Generator and Refinement model, we specify an empty verification to the Refinement model, requiring it to directly refine and improve the generated answer.

### A.10 Limitations and Ethics Statement

We note that even at low temperatures, model performance on benchmarks often exhibits high variance. To address this within our computational constraints, we conducted evaluations on random subsets of test-sets across four seeds. While CommonsenseQA is known to contain many biased or incorrectly labelled questions (Geva et al., 2019), we utilized it in a manner consistent with prior work.

### A.11 Acknowledgments

We thank Aleks Petrov, Dulhan Jayalath, Xingyi Yang, Tala Aljaafari, Markian Rybchuk, Kalyan R, Milind Maiti, Divyansh Garg, and Lewis Hammond for their time and insightful discussions. SM dedicates this work to the memory of his grandmother, Mohini Motwani.

