# OpenReview forum: "MALT: Improving Reasoning with Multi-Agent LLM Training"
_colmweb.org/COLM/2025/Conference — COLM 2025_

### Official Review · Reviewer_s2iA · 2025-05-10

**Rating:** 9
**Confidence:** 4
**Ethics Flag:** 1

**Summary:**

MALT, short for Multi-Agent LLM Training, is a post-training technique for improving the reasoning capabilities of LLMs. The technique builds on the idea that a "division of labor" for reasoning is useful, with one LLM agent responsible for generation of reasoning data, another which does verification/critique, and finally a refiner/editor that takes the critiques and produces a new solution, from which the answer is evaluated. The original element of this paper is the focus on training data instead of focusing on inference time. The authors propose using tree-search to explore reasoning paths, rewarding the paths that lead  to the ground truth answer, and using value iteration for credit assignment. The result can be converted into data for supervised fine-tuning on positive trajectories, or reinforcement learning on the preference pairs (chosen, rejected), for each agent.

The authors demonstrate that the method produces significant gains on standard math/common sense reasoning benchmarks, on the order of 7-16% over single-model baselines, and 3-5% over multi-agent baselines with majority voting. The ablation studies are convincing that the method produces meaningful gains on important benchmarks when used to post-train the Llama-3.1-8B-Instruct model. Of course, as is common in this field, it is an open question whether larger models will see comparable gains from this multi-step method. Nevertheless, I view this paper as an important contribution to the literature, demonstrating a novel technique that shows promise for improving LLM performance on a significant problem.

**Questions To Authors:**

1. I would love to better understand how the authors propose distillation of their trajectories towards training R1-style models. Does this mean selecting the highest value G-V-R path and training with these? Do the authors envision ways to use the lower-value paths to teach the model how to backtrack?

2. Did the authors perform small-scale experiments with $\theta \ne 0.5$ or $n \ne 3$? If so, did the results align with the intuitive explanations provided in the paper?

**Reasons To Accept:**

The paper demonstrates that the MALT technique gives significant gains on challenging reasoning benchmarks. The experiments are compelling, with strong baselines, including multi-agent without MALT and majority voting, as well as standard STaR. The technique makes intuitive sense, with a nice theoretical explanation of MALT as a three-step MDP in the appendix. As stated above, the empirical gains of the method are good, and the techniques make sense, being closely related to how humans solve problems. The authors also provide a full background of years worth of related material, and useful examples of the results of the method. Overall, I found the paper to be accessible and clear, with a strong principled approach and solid empirical results.

**Reasons To Reject:**

I did not find any major reasons to reject this paper, for the reasons stated above. The major points which would strengthen the work, like increasing the dataset sizes and studying the effects of different branching factors and thresholds, are likely too difficult to perform with a standard compute budget. Although I am somewhat skeptical of multi-agent setups in practice, due to the associated complexity and costs, I can fully believe that this method will be useful beyond agentic systems, for example for training thinking models like DeepSeek R1, as the authors allude to.

---

> ### Author Response · Authors · 2025-06-01
>
> Dear Reviewer,
>
> Thank you for your time and valuable feedback! We are glad you found our work novel, promising, and an important contribution. We appreciate the opportunity to further improve our paper and discuss promising future avenues.
>
> We also provide a detailed set of additional experiments on another model (Qwen 2.5 1.5B Base) in our response to Reviewer rgPo. Once again, MALT demonstrates significant performance improvements on GSM8K, and we will expand our experiments and analysis with the Qwen model family to MATH and CSQA for the camera ready version of our paper.
>
> &nbsp;
> ### Questions
> 1. We are very happy to discuss how MALT can be used towards training reasoning models. DeepSeek R1 [1] discusses how high quality cold start data, especially when containing reflection and verification, can be very useful for the reinforcement learning phase. These behaviors and their usefulness for RL, such as backtracking and verification, are also studied in [2]. MALT’s credit assigned reasoning chains, even with our current Generator, Verifier, and Refinement Model setting can be thought of as one reasoning chain for the purposes of generating this cold-start mid-training data. This chain will inherently include verification and backtracking steps due to our multi-agent setup. More specifically, as the reviewer envisions, high-value paths can be used to directly train models to learn the basic cognitive structures and steps required for these behaviors, which RL can then further amplify. Our search based credit assignment strategy yields various benefits to further improve the quality of such data. MALT can allow for the identification of paths, for e.g., where the Generator produces an answer that leads to a low assigned value but still has some correct downstream outputs in the tree. These reasoning chains that converted low value initial generations into high value final outcomes could be specifically used to teach important backtracking and verification behaviors that improve the reasoning performance of the model. Moreover, low and high value paths can be used for preference optimization to further improve the model’s knowledge of these behaviors before the online RL phase to post-train reasoning models. In general, as described in Section 3, MALT can incorporate any meta-strategy, which can then be used as synthetic training data to improve reasoning models. We will provide an even more detailed description of this in the Appendix of our camera ready version.
>
> 2. We performed qualitative experiments early on in our research, where lowering $n$ or $\theta$ reduced the quality of our synthetic training data. With more compute that remains beyond our current scope and resources, both $n$ and $\theta$ can be increased to collect even higher value paths as positive signals while balancing how much training data is generated. In the camera ready version of our paper, we commit to providing a small-scale experiment and analysis that varies the branching factor $n$ to 2 and 5 and increases $\theta$ to 0.7 to evaluate the change in SFT and DPO post-training performance of Generator model.
> Once again, we would like to sincerely thank the reviewer for highlighting the significance and novelty of our work, and engaging in an important discussion about a possible future direction that MALT can be very useful for.
>
> [1] Guo et. al, “DeepSeek-R1: Incentivizing Reasoning Capability in LLMs via Reinforcement Learning”
>
> [2] Gandhi et. al, “Cognitive Behaviors that Enable Self-Improving Reasoners”

---

> > ### Comment · Reviewer_s2iA · 2025-06-05
> >
> > Thank you for your reply! I appreciate the commitment to the additional small-scale experiments. My rating remains unchanged.

---

### Official Review · Reviewer_rgPo · 2025-05-10

**Rating:** 8
**Confidence:** 4
**Ethics Flag:** 1

**Summary:**

This paper introduces MALT, a framework for improving a model's reasoning by jointly training 3 agents to generate, verify, and refine responses. Experiments show that MALT nontrivially improves performance across reasoning-based benchmarks. It serves as a solid proof of concept for the framework, while leaving ample room for future exploration.

**Questions To Authors:**

- Is it possible to repeat the core experiments with any other open source (small) model?
- MALT in general implies that a final model deployed to production requires both MA and MV, as in table 1, MALT without MV performs about the same as inference-only MA, as noted by authors. What happens without even MA, using only the generator, in SA mode?
- What happens as the number of generations for G, V, and R goes up or down?
- Since the generator here uses only SFT and not DPO, doesn't it see significantly less data in training depending on the pass rate of the base policy?
- Is this truly limited to reasoning? Certain justifications are given for negative feedback to improve reasoning, but I see no reason MALT could not also be used to improve soft-verifiable domains (even training against a reward model).

**Reasons To Accept:**

The paper is technically sound, well motivated, and novel. It introduces an interesting framework with many degrees of freedom built in, alongside solid experiments showing the usefulness of the framework.

The authors have addionally provided fair ablations showing the necessity of each part of the framework, and have provided good discussion around possible issues or questions.

**Reasons To Reject:**

My main concerns are around slightly limited experiments - e.g. only 1 model being used for the framework and quirks that come with it. However, I would consider these concerns minor.

- The generator is not trained with DPO and only SFT. A reasonable justification is given in A.10, but this might speak to something more fundamental in the method.
- In table 1, SA STaR doesn't help the model much (GSM8k goes down; CSQA and MATH seem within noise). STaR MA+MV also doesnt see the gains that inference only MA+MV obtains. This is also consistent with the idea that Llama 3.1 has already undergone much training. However, this begs the question of when MALT is useful: is it only useful for such already post-trained models? What happens for other, less trained base policies, or more trained base policies?
- A central feature is the tree-styled exponential branching factor. The authors mention in A.10 that future work includes ablating this factor, but I would prefer some experiments around this.
- The noise levels in table 1 are quite high in some cases, which makes me slightly uncomfortable. However, it's interesting to note that the MALT tests have lower noise in comparison.

---

> ### Author Response · Authors · 2025-06-01
>
> Dear Reviewer,
>
> Thank you for your time and valuable feedback! We are glad you found our paper technically sound, novel, and well motivated. We appreciate the opportunity to further improve our work and answer your questions.
>
> In order to address several of the points raised, we provide a rigorous set of experiments using Qwen-2.5-1.5B Base as the underlying model for MALT and evaluate on GSM8K. We provide our results in the table below along with a detailed discussion.
>
> &nbsp;
> |   Method (on Qwen 2.5-1.5B Base)  | Accuracy on GSM8K |
> | -------- | :-------: |
> | Base Single Agent  | 61.75% &pm; 4.72 |
> | Base Single Agent MV | 65.0% &pm; 3.56 |
> | STaR Single Agent | 63.50% &pm; 1.73 |
> | STaR Single Agent MV | 64.25% &pm; 2.06 |
> | Base Multi Agent | 60.50% &pm; 2.89 |
> | Base Multi Agent MV | 62.25% &pm; 2.22 |
> | STaR Multi Agent | 62.50% &pm; 4.20 |
> | STaR Multi Agent MV | 64.25% &pm; 2.87 |
> | DPO Single Agent | 68.75% &pm; 1.89 |
> | DPO Single Agent MV | 71.25% &pm; 3.59 |
> | MALT without MV | 71.00% &pm; 3.56 |
> | **MALT** | **74.50% &pm; 2.65** |
>
> Here, Single Agent refers to the Generator and MV refers to Majority Voting@3. On the Base model, DPO improves the Generator's performance beyond STaR on the same data, and is done after SFT. MALT surpasses all baselines by a clear margin. Our anonymized inference logs are available at https://shorturl.at/EdtVc
>
> &nbsp;
> ### Weaknesses
> 1. For Llama 3.1 8B Instruct, we provide a justification in Appendix A.10 for not using DPO. Our results with the Qwen Base model show that DPO on a non-overtrained base model for the same data distribution after SFT shows notable improvements (68.75% compared to 63.50% with SFT only).
> 2. Our results on the Qwen Base model show that STaR MA+MV surpasses MA+MV on the Base model. Overall, in the case of our Llama Instruct model, our results across all 6 benchmarks show that MALT can lead to notable performance improvements. Our experiments on a Base model here further substantiate our claims, where the relative improvement is even higher than Instruct models (due to a higher scope for improvement) and shows how MALT can offer substantial benefits in both settings.
> 3. We appreciate the reviewer’s feedback on experiments related to ablating the branching factor. While a detailed experiment remains beyond our current scope due to compute constraints, we commit to providing a small-scale experiment and analysis that varies the branching factor $n$ to 2 and 5 and increases $\theta$ to 0.7 to evaluate the change in SFT and DPO post-training performance of the Qwen Generator model.
> 4. To address the noise levels, we provide a detailed set of experiments with the Qwen model, where MALT on GSM8K shows clear improvements. Therefore, despite some noise (common for small models and because we use 4 seeds), we show improvements for two model families across a range of benchmarks.
>
> &nbsp;
> ### Questions
> 1. Yes, absolutely! We have provided a detailed set of experiments using Qwen 2.5 1.5B Base on GSM8K for this discussion. We commit to expanding these experiments to MATH and CSQA along with a detailed analysis in our camera ready version.
> 2. For the Instruct model, using the Generator in SA mode is equivalent to our STaR SA baseline. For the Base model, it is equivalent to our DPO baseline.
> 3. We hypothesize that if the number of G, V, and R samples go up at training, this would allow for more synthetic multi-agent data to be generated. While this would require additional computational resources, we expect more synthetic data to be useful and will evaluate with a branching factor of 5 in our camera ready version, as described above.
> 4. In the case of Llama 3.1 8B Instruct, the Generator is trained with SFT only. However, it sees roughly the same amount of data (i.e. training questions), only without the incorrect preference example which was used for DPO. After the credit assigned synthetic data generation process, all models had roughly a similar number of training examples. MALT therefore remains an important source of producing multi-agent role-specific post-training data, regardless of whether only SFT or both SFT and DPO are used.
> 5. We completely agree. MALT can potentially be extended to soft reasoning domains, where multi-agent setups can improve performance with the guidance of an outcome based Generative (Generalist) Reward Model [1] or similar soft verification function. MALT provides an extensible method for multi-agent setups to learn via credit assignment and post-training on any domain.
>
> We sincerely appreciate the reviewer’s feedback and have conducted important additional experiments to answer questions. We commit to providing a detailed analysis of our results with Qwen 2.5 1.5B Base on GSM8K, MATH, and CSQA, along with experiments related to modifying the branching factor. We would really appreciate it if the reviewer would consider raising their score further.
>
> [1] Lie et. al, “Inference-Time Scaling for Generalist Reward Modeling”

---

> > ### Comment · Reviewer_rgPo · 2025-06-06
> >
> > Thanks for the additional experiments and the in-depth replies! This seems like a good area to pursue further. Having another base model helps alleviate concerns around noise.
> >
> > Maybe a final stylistic suggestion (more of an opinion) would be to slim down the introduction slightly and put more emphasis on the novelty in having trained specialized agents. I think this would help make novelty in the work clearer.
> >
> > Additionally, maybe the tables could be slimmed down by splitting SA/MA and SV/MV into other tables, highlighting only the MA/MV setup in the main text. I understand the full set of numbers is important, but maybe it makes highlighting results difficult. Again, probably more of my own opinion - no need to change if you disagree.
> >
> > I'll move the confidence in my review to a 4 with an 8 overall, I will think further on the overall score before the discussion period ends and will read thoughts from the other reviewers.

---

### Official Review · Reviewer_Shaf · 2025-05-18

**Rating:** 4
**Confidence:** 4
**Ethics Flag:** 1

**Summary:**

This paper proposes MALT, a post-training method that assigns reasoning sub-tasks to three specialized agents: a generator, verifier, and refiner. It introduces a search-tree-based data generation process with value iteration for automatic credit assignment, enabling supervised fine-tuning and DPO without intermediate human labels. Experiments on GSM8K, MATH, and CSQA show performance gains over both single-agent and inference-only multi-agent baselines.

While the system is well-engineered and results are promising, the overall novelty is moderate. The method largely builds upon existing work such as STaR, DPO, and multi-agent debate frameworks, integrating them into a coherent pipeline rather than proposing fundamentally new techniques.

**Reasons To Accept:**

1/ The method yields decent performance improvements across reasoning benchmarks and generalizes well to symbolic and out-of-domain tasks.
2/ The value iteration credit assignment enables automated training of multi-agent systems without requiring human annotations or oracle models.

**Reasons To Reject:**

1/ The core components—multi-agent design, DPO, synthetic search-based data—are all derived from prior work (e.g., STaR, AutoGen, Debate Agents), with MALT functioning as a well-engineered workflow rather than a methodology. Without majority voting, the improvement is also marginal compared to Star baseline (within std).

---

> ### Author Response · Authors · 2025-06-01
>
> Dear Reviewer,
>
> Thank you for your review and for noting that MALT leads to performance improvements and that our value-iteration credit assignment removes the need for human labels or oracle models. We address the concerns raised in your review below:
>
> 1. Novelty: we would like to clarify that MALT is more than reused parts.
>
> * Unlike AutoGen or Debate Agents, and multi-agent debate frameworks, which repurpose a single model with different prompts, we learn specialized agents with dedicated functions. More specifically, we do not use these frameworks for any part of our method other than drawing inspiration from multi-agent LLM setups.
> * We show how our method does credit assignment without extra supervision, relying only on outcome rewards for training the agents. We also provide a theoretical justification for the same.
> * The nontrivial combination of the above with SFT + DPO yields a new post-training paradigm for complex multi-agent reasoning and demonstrates important empirical improvements.
>
> 2. Majority voting fairness: all baselines, including every STaR variant, were evaluated with the same MV budget. In this fair setting, MALT (which includes MV) outperforms baselines on six benchmarks and even generalizes to unseen tasks (eg, AIME, MMLU-Pro, GSM-Symbolic). We also provide a detailed discussion of why we use MV and how it improves MALT significantly compared to other baselines in Section 5.2 (Self-consistency). Moreover, in our response to reviewer rgPo, we provide further evaluations on another model family, where MALT clearly surpasses all other baselines.
>
> We would also like to highlight the novelty and importance of our paper as recognized by reviewers rgPo and s2iA, who state that it is “novel, an important contribution, technically sound, well-motivated, has many degrees of freedom built in, and provides significant gains on reasoning benchmarks”.
>
> We hope this clarifies why we believe MALT is a novel training paradigm rather than an engineering tweak, and we appreciate any further feedback you may have. We would be grateful if, given the information and clarifications here and in the paper, the reviewer might consider reevaluating and increasing their score.

---

### Decision · Program_Chairs · 2025-07-08

**Decision:**

Accept

**Comment:**

The paper proposes a new training paradigm rather than playing at inference time, it learns 3 specialized agents via a tree‑based, automatically labeled dataset and combines SFT with preference optimization. The novelty is integration into a unified, training‑time pipeline with principled credit assignment is, in my judgment, a substantive methodological advance that unlocks clear performance gains and good cross‑task generalization, although as reviewers pointed out they are not individually novel components.
The additional Qwen experiments alleviate the single‑model concern and demonstrate scalability to a weaker base policy. Remaining weaknesses compute cost of n³ sampling and lack of very‑large‑model results are genuine but do not, in my view, outweigh the contribution. Reviewer Shaf’s novelty objection is noted, and the rebuttal addresses fairness/ablation questions satisfactorily.

I agree that MALT’s novelty lies in how these components are orchestrated into a coherent, trainable multi-agent system with credit assignment. This is not trivial, prior work has not used value iteration over a reasoning search tree to train separate agents via outcome-based feedback. There is also a strong generalization to out-of-domain tasks on AIME, MMLU-Pro.

highly suggest the authors to include:
1. The promised branching‑factor ablation and expanded Qwen results.
2. Introduction foregrounds the training novelty more succinctly (per rgPo).
3. Compute/efficiency discussion is strengthened (runtime, GPU hours, scaling advice).

[Automatically added comment] At least one review was discounted during the decision process due to quality]